# Seroprevalence of the Hepatitis E Virus in Indigenous and Non-Indigenous Communities from the Brazilian Amazon Basin

**DOI:** 10.3390/microorganisms12020365

**Published:** 2024-02-10

**Authors:** Mariana Pinheiro Alves Vasconcelos, Jaqueline Mendes de Oliveira, Juan Camilo Sánchez-Arcila, Sarah Castro Faria, Moreno Magalhães Rodrigues, Daiana Perce-da-Silva, Joffre Rezende-Neto, Marcelo Alves Pinto, Marilza Maia-Herzog, Dalma Maria Banic, Joseli Oliveira-Ferreira

**Affiliations:** 1Laboratório de Imunoparasitologia, Instituto Oswaldo Cruz—FIOCRUZ/IOC, Rio de Janeiro 21045-900, Brazil; marianapvasconcelos@hotmail.com (M.P.A.V.); sanchezarcila.jc@gmail.com (J.C.S.-A.); 2Centro de Medicina Tropical de Rondônia—CEMETRON, Porto Velho 76812-329, Brazil; 3Laboratório de Desenvolvimento Tecnológico em Virologia, Instituto Oswaldo Cruz, Fiocruz, Rio de Janeiro 21045-900, Brazil; jackie@ioc.fiocruz.br (J.M.d.O.); sarahsalvadorms@gmail.com (S.C.F.); marcelop@ioc.fiocruz.br (M.A.P.); 4Fundação Oswaldo Cruz—FIOCRUZ, Porto Velho 76812-245, Brazil; moreno.rodrigues@fiocruz.br; 5Laboratório de Imunologia Clínica, Instituto Oswaldo Cruz—FIOCRUZ/IOC, Rio de Janeiro 21045-900, Brazil; daiana_perce@hotmail.com (D.P.-d.-S.); banic@ioc.fiocruz.br (D.M.B.); 6Instituto Joffre Rezende, Goiânia 74085-100, Brazil; drjoffre.neto@gmail.com; 7Laboratório de Referência Nacional em Simulídeos, Oncocercose e Mansonelose, Coleção de Simulídeos do Instituto Oswaldo Cruz—FIOCRUZ/IOC, Rio de Janeiro 21045-900, Brazil; iocherzog@gmail.com

**Keywords:** hepatitis E virus (HEV), hepatitis E virus antibodies (anti-HEV), hepatitis E seroprevalence, Brazilian Amazon Region, Brazilian North Region, indigenous

## Abstract

Hepatitis E virus (HEV) infection is a common cause of acute viral hepatitis in tropical regions. In Brazil, HEV G3 is the only genotype detected to date. Reports on HEV prevalence are heterogeneous. We aimed to compare the prevalence of anti-HEV among three populations living in the Brazilian Amazon basin. Two cross-sectional studies were conducted in urban, rural, and Yanomami indigenous areas. Plasma samples from 428 indigenous and 383 non-indigenous subjects were tested for anti-HEV IgG using enzyme-linked immunosorbent assays. The overall prevalence of anti-HEV was 6.8% (95%CI: 5.25–8.72), with 2.8% (12/428) found in the Yanomami areas, 3% (3/101) in an urban area, and 14.2% (40/282) in a rural area. Multivariate logistic analysis indicated that patients aged 31–45 years or ≥46 years are more likely to present anti-HEV positivity, with a respective aOR of 2.76 (95%CI: 1.09–7.5) and 4.27 (95%CI: 1.58–12.35). Furthermore, residence in a rural area (aOR: 7.67; 95%CI: 2.50–33.67) represents a relevant risk factor for HEV infection. Additional studies detecting HEV RNA in fecal samples from both humans and potential animal reservoirs are necessary to comprehensively identify risk factors associated with HEV exposure.

## 1. Introduction

The Hepatitis E virus (HEV), classified under the *Paslahepevirus balayani* species (Paslahepevirus genus) of the Hepeviridae family, is a single-stranded positive-sense RNA virus that infects both humans and a wide range of domestic and wild animal species [1]. Members of the *Paslahepevirus balayani* species are classified into eight genotypes. HEV G1 and G2 infect only humans, while HEV G3 and G4 can infect both humans and diverse animal species, with *Suidae* species being the main reservoir. Within the HEV G3, a separate branch represents HEV found in rabbits (HEV genotype 3ra), which also includes a closely related human strain [2,3]. Two other HEV genotypes, G5 and G6, have exclusively been described in wild boars from Japan [4].

HEV genotypes 7 and 8 were first discovered in dromedary camels and later in a patient with chronic post-transplant hepatitis [5,6]. Zoonosis via rodent-borne HEV belonging to the *Rocahepevirus ratti* species was recently demonstrated [7]. HEV is mainly transmitted by the fecal–oral route from contaminated water or food or by contact with infected animals or their environments [8]. Most HEV infections lead to self-limited acute hepatitis. However, progression to chronic hepatitis has primarily been reported in immunosuppressed recipients of solid organ transplants [8,9,10]. Fulminant hepatitis is rare, with the majority of reported cases occurring in pregnant women residing in highly endemic areas where HEV genotype 1 is the most prevalent [9]. 

The World Health Organization estimates approximately 20 million cases of HEV infections worldwide each year, 3.3 million of which present symptoms, resulting in 44,000 deaths annually [8]. HEV infections are predominantly prevalent in underdeveloped countries and regions with limited access to clean water and basic sanitation, such as Africa and Asia [8,9,11]. In these regions, both HEV G1 and G2 have been associated with outbreaks. In Brazil and other South American countries, G3 is most frequently detected, while HEV G1 has only been reported in Venezuela and Uruguay [12,13,14]. 

HEV G3 has been detected in humans, animal hosts, and contaminated environmental sources. This genotype is commonly associated with animal contact and the consumption of contaminated food, such as undercooked or raw meat [15,16]. The epidemiology of HEV infection in Brazil is poorly described, as it is not considered a notifiable disease. HEV infections often go undiagnosed due to the possibility of mild or asymptomatic cases going unreported or unnoticed. 

Hepatitis E virus (HEV) has the potential to affect a wide range of populations, including individuals residing in urban areas. However, indigenous, rural, and riverine communities may face a significantly higher risk due to exposure to HEV risk factors. The first serological evidence of HEV infection in rural and indigenous communities in South America emerged in the 1990s in Venezuela [17], Chile [18], and Bolivia [19]. During this period, anti-HEV prevalence reportedly ranged from 3.9% to 7.3% in rural areas and from 9.7% to 20.1% in indigenous villages. 

In Brazil, several studies have reported the prevalence of anti-HEV as ranging between 2.1% and 8.4% in rural regions [20,21,22,23]. Vitral et al. (2014) found a higher prevalence of 12.9% among individuals residing in a rural settlement in Granada, Acre, situated in the western Amazon basin [24,25]. 

Previous studies have reported HEV seroprevalence rates of 6% among gold miners [26], 3.3% in the general population of the southern Amazon basin [23], 4.5% in children aged 2 to 9 years old in an Amazon community in the state of Mato Grosso [27], and 4% in riverine communities residing in the western Amazon basin [28]. However, the highest HEV seroprevalence has been reported in urban areas in the southern region of Brazil. In the state of Rio Grande do Sul, Pandolfi et al. [29] reported a 40.25% prevalence of anti-HEV antibodies among blood donors. Another study confirmed high HEV seroprevalence in the general populations of three municipalities in Rio Grande do Sul, reaching 55.4%, 57.4%, and 65.5% [30].

To date, a limited number of studies have attempted to determine the seroprevalence of HEV in indigenous populations across South America. Recently, Villar et al. (2020) reported a remarkably low anti-HEV prevalence (0.2%) in the Apinajé indigenous population of the western Brazilian Amazon [31]. Other studies have reported anti-HEV seroprevalence rates of 5.6% in indigenous communities in Argentina [32] and 1.9% in Suriname [33].

Despite not being listed as a neglected tropical disease by the World Health Organization (WHO), HEV should be considered a neglected tropical disease due to its burden, lack of attention from public health, research and clinical communities, and limited treatment and control options [34]. A recent analysis revealed that very limited investment has been made in Hepatitis E research: From 2008 to 2017, research funding was 30 times lower than funding for other neglected diseases, such as cholera and typhoid fever [35].

In light of this scenario, the present study aimed to investigate the seroprevalence of anti-HEV antibodies and the potential risk factors associated with HEV exposure in both indigenous and non-indigenous populations residing in the Brazilian Amazon basin. 

## 2. Materials and Methods

### 2.1. Study Area and Population

Two cross-sectional studies were conducted in the Brazilian Amazon Basin. One examined rural and urban populations in Rondônia state (2010–2011) while the other assessed a community in the Yanomami indigenous territory in the state of Amazonas (2015). Both studies were originally designed to investigate coinfections of malaria and intestinal parasites [36,37]. The extremely high prevalence of intestinal parasites found in all three populations investigated confirmed the unique environmental, cultural, and socioeconomic factors contributing to poor sanitary conditions in the region. For the purposes of this study, previously collected samples were subsequently used to assess the incidence of HEV across all three populations. 

Rondônia, located in the Western Amazon Basin of Brazil, borders Bolivia and the Brazilian states of Amazonas, Acre, and Mato Grosso (Figure 1). Porto Velho, the capital of Rondônia, covers an area of 34,090,952 km^2^ (8°45′45″ S 63°53′12″ W) and has an estimated population of 460,413 inhabitants. Individuals from both rural and urban areas of the city of Porto Velho were included. The rural population of Porto Velho that was studied is located near the Joana D’Arc settlement; their primary economic activity is agriculture; livestock agriculture, including the production of bovine milk and cheese; and raising small animals, such as poultry and pigs. This area is characterized by poor housing conditions, a lack of sanitary infrastructure, and predatory forest invasion, all of which have contributed to high levels of morbidity associated with malaria and other poverty-associated diseases. By contrast, the urban area of Porto Velho is densely populated and has urban infrastructure, such as water supply, basic sanitation, paved streets, commercial and residential buildings, public and private services, and public transport, among others. A more complete description of the studied areas has been previously reported [37]. 

The Yanomami, a semi-isolated Amazonian indigenous people, are the native inhabitants of an approximately 74,000 square mile territory straddling the Brazilian–Venezuelan border of the states of Roraima and Amazonas. The Marari region, where one of the 37 Yanomami public health units is located, provides primary health assistance to four indigenous villages within a remote area of the Amazon rainforest surrounded by high mountains, which is only accessible by small plane or boat. Samples taken from these four Yanomami villages in Marari were tested for anti-HEV antibodies. The sample size comprised 126 samples from the Castanha/Ahima village (location: 1°11′39.3″ N 64°48′39.2″ W), 105 samples from Gasolina (location: 1°08′58.4″ N 64°48′40.3″ W), 121 samples from Taibrapa (location: 1°10′46.8″ N 64°47′52.1″ W), and 78 samples from the Alapusi village (location: 1°11′31.8″ N 64°51′30.2″ W). All of the Yanomami communities in Marari continue to maintain traditional livelihoods. Hunting is an important source of protein for the Yanomami, and several species of animals complement their plant-based diet, including monkeys, deer, tapirs, poultry (birds), and armadillos. All four villages lack toilets, and river water is used for bathing, drinking, and cooking (Figure 1). 

### 2.2. Blood Sample Collection 

Blood samples (5 mL) were previously collected from each participant via venipuncture in 5 mL tubes containing EDTA anticoagulant (BD Vacutainer^®^, Franklin Lakes, NJ, USA). All samples were processed in the field to obtain plasma, which was then stored in cryotubes in liquid nitrogen—LN2. Plasma samples were eventually sent to the Laboratory of Immunoparasitology at the Oswaldo Cruz Institute (Fiocruz, Rio de Janeiro, Brazil), where they were maintained at −20 °C. Cryopreserved plasma samples were finally sent to the Fiocruz Laboratory of Technological Development in Virology for anti-HEV serological analysis.

### 2.3. Serological Analysis 

Two commercial enzyme-linked immunosorbent assays (ELISAs) were used to test plasma samples for anti-HEV IgG. The bioElisa HEV IgG (Biokit, Barcelona, Spain) was first used for the urban and rural populations, while the recomWell HEV IgG ELISA kit (Mikrogen Diagnostik, Neuried, Germany) was later used in the Yanomami cross-sectional study because the Bioelisa (Biokit) assay was no longer commercially available at the time. All tests were performed in accordance with the manufacturer’s instructions. The interpretation of results obtained from the two serological surveys with different cut-off (CO) calculations was standardized as follows: Results were classified as positive or negative by comparing the “signal to cut-off ratio”, i.e., the ratio of the optical density (OD) reading (at 450 nanometers) of each sample to the cut-off value defined for each assay. Samples with an OD/CO ratio ≥ 1.2 were considered positive, while those with an OD/CO ratio < 0.9 were considered negative. Borderline results (0.9 > OD/CO ≤ 1.2) were not considered in the analysis. A total of 18 individuals were excluded from the statistical analysis due to borderline or indeterminate anti-HEV results. Specifically, six individuals resided in the urban area of Porto Velho, ten were from the rural area of Porto Velho, and two were from the Yanomami villages.

### 2.4. Ethical Approval

The recruitment of participants in the Yanomami villages was aided by a bilingual interpreter who explained to the indigenous leaders and/or representatives the purpose of the study and the procedures to be carried out. Verbal and written informed consent for each participant was obtained; in the case of minors, their guardians provided consent prior to inclusion in the study. Local representatives supervised all fieldwork. The ethical and methodological aspects of this research were approved by the National Research Ethics Committee (CONEP, under protocol No. 16.907). 

In the urban and rural communities of Rondônia, ethical approval was obtained from the local Institutional Review Board of the Oswaldo Cruz Foundation (CEP/FIOCRUZ, 492/08). All study participants provided written informed consent prior to study inclusion, and they were subsequently submitted to physical examinations by a physician.

### 2.5. Statistical Analysis

Categorical variables were described using n and the proportion of patients in each group, while numerical variables were described using medians and the interquartile range (IQR). To compute the unadjusted odds ratio (OR), we performed bivariate analyses using Anti-HEV results (0 = negative and 1 = positive) as the outcome for each population. Due to missing results in one or more variables, 27 individuals were removed from the modeling process. To compute adjusted OR (aOR), a multivariate generalized linear model using Anti-HEV as a response variable (Y) was adjusted to a binomial distribution. In this case, let Y*_i_* be the Anti-HEV result for a given patient, where *i* = 1, …, 801, and the linear predictor could be expressed as 6.78% (95%CI: 5.25–8.72).
$$g\left(\mu_{i}\right)=\beta_{0}+\beta_{1}\times Age+\beta_{2}\times sex+\beta_{3}\times Area+\beta_{4}\times Occupation+\beta_{5}\times Region$$

In this case, *g* is the link function that, in the case of the binomial model, assumes the following:$$log\left(\frac{\mu_{i}}{1-\mu_{i}}\right)$$

Then, aOR is calculated by an exponential (*β_i_*). All analyses were performed using R 4.3.0 statistical software (R Development Core Team, 2023). 

## 3. Results

### 3.1. Anti-HEV Detection

Two cross-sectional studies were previously conducted in indigenous villages (2015) and urban and rural areas (2010–2011). A total of 811 plasma samples were collected from 428 indigenous Yanomami and 383 were collected from non-indigenous urban (n = 101) and rural residents (n = 282). Plasma samples were tested for anti-HEV IgG using two ELISA assays. The overall prevalence of anti-HEV among all studied subjects was 6.78% (95%CI: 5.25–8.72).

Of the 101 individuals from the urban area of Porto Velho, a total of 3 individuals tested positive for anti-HEV antibodies, resulting in a prevalence of 3%. The median age of this population was 31 years (IQR: 23–44 years), with confirmed cases comprising two females and one male aged 30 years or older. While no statistically significant differences were seen in either age (*p* = 0.39) or gender (*p* = 0.57), it must be acknowledged that the overall sample size was quite small. Of the three individuals who tested positive for anti-HEV antibodies, two were healthcare professionals. In the urban area, only 3.9% (4/101) of the individuals studied reported working with activities related to agriculture or livestock production, yet none of them tested positive. All individuals with positivity for anti-HEV antibodies were born in the northern region of Brazil.

In the rural area of Porto Velho, the prevalence of individuals who are positive for anti-HEV antibodies was 14.2% (40/282). The median age of these subjects was 30 years (IQR: 15–42 years) and over 60% of the cases were aged 30 years or older (*p* = 0.01). Although females accounted for 55.9% of the cases, this result was not statistically significant (*p* = 0.33). Considering the nearly one-third of this rural population works in agriculture and/or animal husbandry, 12.0% (10/83) of these individuals were positive for anti-HEV antibodies. Of the individuals who reported “homemaker” as their profession (11/56), all were female and the prevalence of anti-HEV antibodies was positive for 19.6%. 

Considering the individuals who were not born in the North region, the median time of residence in the region was over 20 years, regardless of location (i.e., rural or urban area).

The Yanomami indigenous population had an overall anti-HEV prevalence of 2.8% (12/428). Of the anti-HEV-positive individuals, 10/12 were male (83%, *p* = 0.009) with a median age of 26 years (IQR: 17–42 years). Notably, 50% of the anti-HEV-positive individuals were aged over 45 years. Of the 12 anti-HEV-positive individuals, 6 (5.8%) were from the village of Gasolina, 4 (3.3%) were from Taibrapa, and 2 (2.6%) were from Alapusi village. HEV antibodies were not detected in any of the inhabitants of the Castanha/Ahima village. Nonetheless, differences in HEV prevalence between the villages were not statistically significant (Table 1).

### 3.2. Odds Ratio and Adjusted Odds Ratio 

Anti-HEV IgG was detected in 55 (approximately 6.7%) of the 811 individuals studied. An exploratory bivariate analysis indicated a higher likelihood of a positive anti-HEV IgG test for individuals who live in rural areas (aOR = 5.14; 95%CI 1.81–22.4) compared to those who live in urban areas. Furthermore, we found that those who live in the North Region were less likely to have a positive anti-HEV IgG (0.45; 0.24–0.89) compared with those who are from other regions (Table 2). In the multivariate analysis, some groups were more likely to have a positive anti-HEV IgG test, namely, young adults (31 and 45 years old) (aOR = 2.76; 95%CI 1.09–7.50) or middle-aged adults >= 46 years old (4.27; 1.58–12.35) and those who live in rural areas (7.67; 2.50–33.67). The aOR indicating no association between living in the North Region comprised individuals who either have been a farmer or had a positive anti-HEV test (aOR = 1.38; 0.63–3.1) (aOR = 0.44; 0.17–1.06) (Table 2).

### 3.3. Multivariate Logistic Analysis

The results of multivariate logistic analyses indicated that patients aged 31–45 years or ≥46 years are more likely to exhibit anti-HEV positivity, with respective aOR values of, 2.76 (95%CI: 1.09–7.5), and 4.27 (95%CI: 1.58–12.35). Furthermore, residence in the rural area (aOR: 7.67; 95%CI: 2.50–33.67) was found to represent a relevant risk factor for HEV infection (Figure 2).

## 4. Discussion

Hepatitis E virus infection is a common cause of acute viral hepatitis in tropical areas, with HEV G1 being the most prevalent genotype, followed by HEV G2, which is only found in regions highly endemic for HEV [38]. In developing regions, such as Africa, Asia, and South America, HEV infections remain underdiagnosed due to a lack of surveillance and diagnostic efforts, leading to unknown disease prevalence [39]. HEV G3, which is generally associated with asymptomatic infection, is the only genotype that has been detected in Brazil to date. While reports on the prevalence of HEV antibodies are heterogeneous, the data show an increasing trend from the northern to southern regions of Brazil. The literature contains scarce information on HEV seroprevalence and risk factors for HEV exposure in the Amazon region, particularly among indigenous populations.

The present study found a prevalence of 3.0% in the urban population of Porto Velho, which stands in contrast to another study investigating anti-HEV antibody prevalence in an urban population of Tocantins (Amazon region). These authors reported that none of the 175 individuals tested positive for HEV [31]. Studies on the general population and blood donors from the southern region of Brazil indicated some of the highest rates of HEV seroprevalence in the country, ranging between 40% and 65% [29,30,40]. We found an HEV seroprevalence of 14.2% in the rural population of Rondônia, which is comparable to another finding of 12.9% in a rural settlement in the state of Acre in the Brazilian Amazon region [24]. In these Brazilian rural communities, most individuals are actively involved in farming or are homemakers. In the rural population studied herein, individuals who were homemakers or worked in agriculture/husbandry presented prevalence rates of 19.6% and 12.0%, respectively [25,41,42]. 

Although this study did not attempt to investigate the route of HEV transmission, it seems that the pattern of infection in the rural community investigated is similar to that observed in developed countries for HEV G3, in which most cases can be attributed to contaminated food, consumption of contaminated shellfish, undercooked pork or wild game and direct exposure to pigs [43]. It is worth noting that HEV G3 has been detected in Brazilian patients with acute hepatitis, as well as swine herds in the state of Pará, located in the eastern Brazilian Amazon region [25,44]. 

In the Yanomami indigenous population investigated herein, the prevalence of HEV IgG antibodies was 2.8%, higher than in an indigenous population of Tocantins (0.2%) but lower than the rates observed in a Guaraní indigenous population of northern Argentina (5.6%) and an indigenous population of Malaysia (5.9%) [31,32,45]. Very low anti-HEV prevalence (0.19%) was recently reported in an Apinagé indigenous community in the western Amazon region [31]. 

Indigenous populations often live in poor sanitary conditions and do not have access to potable water. A recent study by our group showed a 100% prevalence of intestinal parasite infections among the Yanomami population [36]. However, despite this finding, the low prevalence (2.8%) of HEV antibodies in this population was unexpected, as both intestinal parasite infection and HEV share similar routes of transmission. In contrast, studies conducted in rural populations in the southern and southeastern regions of Brazil have reported very high prevalence rates of anti-HEV antibodies (up to 65% and 24%, respectively), which is likely associated with foodborne HEV-G3 transmission [29,30,46]. Although the circulation of HEV among domesticated animals and wild swine or other potential animal reservoirs has not been studied in the Yanomami indigenous territory, it is critical to emphasize that the Yanomami do not habitually raise animals or drink their milk. However, there is the potential risk of exposure to HEV through direct contact with wild animals during hunting activities, which are mainly carried out by men. It is worth noting that 83% of the HEV IgG-positive individuals among the Yanomami population were male, with a median age of 26 years. The higher prevalence of disease in men has been attributed to a greater number of behavioral risk factors compared to women [47]. 

The characterization of hepatitis E virus is not only limited to fecal–oral transmission associated with poor sanitary conditions, but it must also be considered as a zoonotic disease linked to direct contact with animals. Thus, HEV can be classified as an occupational disease. The results of our study demonstrate that individuals residing in a rural area, most of whom were farmers and homemakers, presented the highest prevalence of HEV antibodies (14.2%) of the three groups investigated. Previous surveys in the Brazilian Amazon reported that anti-HEV IgG antibodies were detected in 12.9% of individuals from a rural area in Acre compared to 4% in riverine communities from the Brazilian western Amazon region [24].

It is important to note that since the Amazon is one of the most prevalent regions for hepatitis B and D infection worldwide, the occurrence of acute HEV infection in individuals with pre-existing liver disease may result in worse outcomes [48]. 

Our study corroborates findings reported in previous studies indicating a low overall prevalence of anti-HEV antibodies in the Brazilian North region, despite a high incidence of hepatitis A virus (HAV) infection and other enterically transmitted viral and parasitic infections [24]. Although studies have investigated the seroprevalence of HEV in the Amazon region [23,28,31], this is the first report focused on the presently studied areas. The data presented herein are novel and provide important epidemiological information, particularly regarding the circulation of the virus in remote areas, such as the Yanomami indigenous territory. However, it is important to acknowledge certain limitations, particularly the language barrier faced by the Yanomami indigenous population, which hindered the investigator’s obtainment of relevant clinical and epidemiological information. It is important to consider another limitation of our study, as we used two different ELISA kits to detect HEV IgG antibodies. Specifically, we used the recomWell ELISA (Mikrogen) kit for the Yanomami cross-sectional study, as the Bioelisa (Biokit) assay used in our previous study of urban and rural populations was no longer commercially available. It is worth noting that the most commonly used diagnostic tests are Mikrogen and Wantai. Bendall et al. (2010) argued that serological surveys using non-Wantai assays may underestimate HEV-3 antibody prevalence [49]. However, Almeida e Araújo et al. (2020) found comparable sensitivity and specificity between these commercial ELISA kits [46]. Furthermore, our former study, which used the Biokit ELISA, supported data from other authors regarding the seroprevalence of HEV IgG antibodies. Further, studies targeting the detection of HEV RNA in fecal samples from both humans and potential animal reservoirs will be necessary to comprehensively evaluate the risk factors associated with HEV exposure, especially in rural areas in the Amazon region.

## 5. Conclusions

The present study demonstrated the presence of anti-HEV IgG in urban, rural, and indigenous populations in Rondônia and Amazon states, which is suggestive of recent or past HEV circulation in these populations. Therefore, in order to better characterize the epidemiology and impact of HEV in these areas of the Amazon region, further studies should be conducted, targeting the HEV genome in stool samples and assessing the zoonotic potential of HEV infection in these areas, including in wild and domestic animal populations.

## Figures and Tables

**Figure 1 microorganisms-12-00365-f001:**
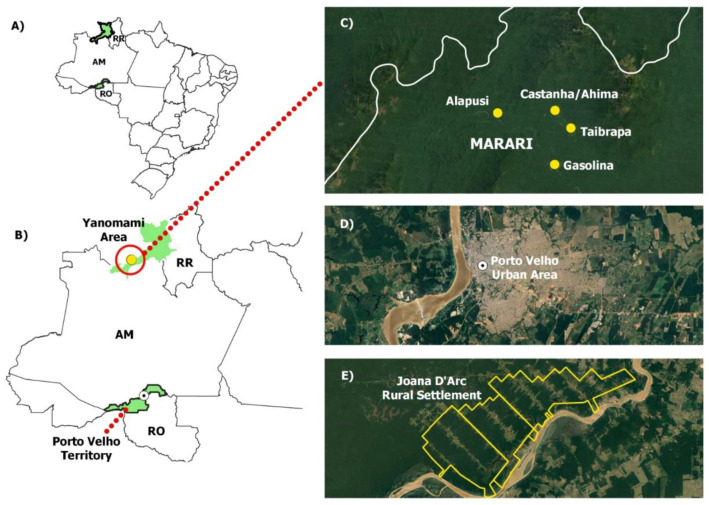
Study areas. (**A**) The map details the location of the studied areas in the Western Amazon region of Brazil. (**B**,**C**) The Marari region in the Yanomami indigenous area. (**B**,**D**) The urban area of the municipality of Porto Velho (Rondônia). (**E**) Rural area on the outskirts of Porto Velho (Joana D’Arc Settlement).

**Figure 2 microorganisms-12-00365-f002:**
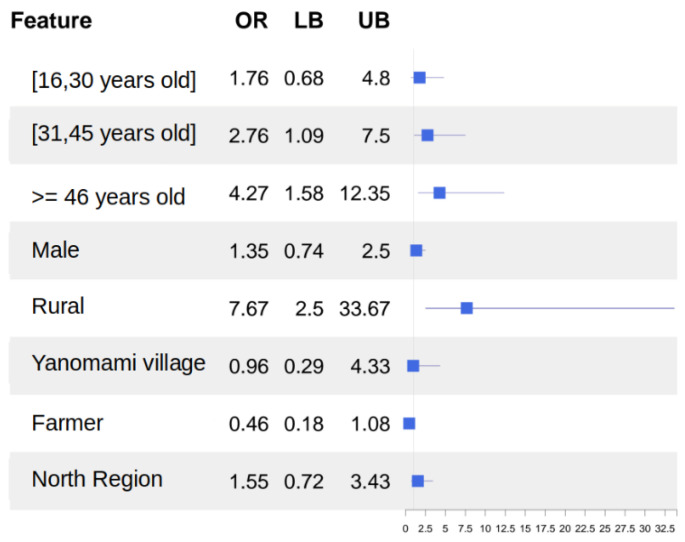
Multivariate logistic analysis of demographic characteristics and HEV risk factors.

**Table 1 microorganisms-12-00365-t001:** Absolute (n) and relative (%) frequency of HEV antibodies and demographic variables in the Brazilian Amazon basin.

	Urban		Rural		Indigenous	
	n = 101		n = 282		n = 428	
	Pos/n (%)	*p*	Pos/n (%)	*p*	Pos/n (%)	*p*
HEV	3/101 (3.0)		40/282 (14.2)		12/428 (2.8)	
Sex		*0.575*		*0.33*		*0.009*
Female	2/42 (4.7)		20/130 (15.38)		2/246 (0.8)	
Male	1/59 (1.7)		18/149 (12.0)		10/180 (5.5)	
Missing	-		3		2	
Age (years)		*0.397*		*0.011*		*0.162*
[0–15]	0/4 (0)		6/76 (7.8)		2/95 (2.1)	
[16–30]	0/46 (0)		9/64 (14.06)		3/149(2.0)	
[31–45]	2/30 (6.7)		15/88 (17.0)		1/89 (1.1)	
45+	1/21 (4.8)		8/51 (15.7)		6/89 (6.7)	
Missing	-		3		6	
Place of birth		*1*		*0.37*		
Midwest	0/2 (0.0)		1/19 (5.2)		-	-
Northeast	0/7 (0.0)		2/24 (8.3)		-	-
North	3/80 (3.7)		23/173 (13.3)		12/428 (2.8)	
Southeast	0/5 (0.0)		6/27 (22.2)		-	
South	0/4 (0.0)		5/33 (15.1)		-	
Missing	3		6		-	
Indigenous Village						*0.066*
Alapusi	-		-		2/76 (2.6)	
Castanha/Ahima	-		-		0/126 (0.0)	
Gasolina	-		-		6/97 (6.2)	
Taibrapa	-		-		4/117 (3.4)	
Occupation		*0.47*		*0.335*		
Farmer	0/4 (0.0)		10/83 (12.0)		-	
Health worker	2/20 (10.0)		1/7 (14.2)		-	
Homemaker	0/7 (0.0)		11/56 (19.6)		-	
Other	1/42 (2.4)		1/24 (4.2)		12/428 (2.8)	
Missing	28		112		-	

**Table 2 microorganisms-12-00365-t002:** Results of the adjusted logistic regression models analyzing the presence of anti-HEV antibodies in individuals from the Amazon Basin region (Brazil).

	Anti-HEV Negative	Anti-HEV Positive	OR	*p*-Value
Age (years)				0.103
[0–15]	167 (22.3%)	8 (15.1%)	Ref.	
[16–30]	247 (33.0%)	12 (22.6%)	1.01 [0.40; 2.66]	
[31–45]	189 (25.2%)	18 (34.0%)	1.96 [0.85; 4.95]	
45+	146 (19.5%)	15 (28.3%)	2.12 [0.89; 5.47]	
Sex				0.396
Female	394 (52.3%)	24 (45.3%)	Ref.	
Male	359 (47.7%)	29 (54.7%)	1.32 [0.76; 2.34]	
Area of residence				<0.001
Urban	98 (13.0%)	3 (5.45%)	Ref.	
Rural	242 (32.0%)	40 (72.7%)	5.14 [1.81; 22.4]	
Yanomami territory	416 (55.0%)	12 (21.8%)	0.91 [0.28; 4.24]	
Occupation				0.10
Non-Farmer	679 (89.8%)	45 (81.8%)	Ref.	
Farmer	77 (10.2%)	10 (18.2%)	1.98 [0.90; 3.95]	
Place of birth				0.023
Non-North region	107 (14.3%)	14 (26.9%)	Ref.	
North Region	643 (85.7%)	38 (73.1%)	0.45 [0.24; 0.89]	

## Data Availability

The datasets generated and/or analyzed during the current study are available from the corresponding author upon reasonable request.

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
