# Peer review of "Seroprevalence of the Hepatitis E Virus in Indigenous and Non-Indigenous Communities from the Brazilian Amazon Basin"

_microorganisms, 2024, doi:10.3390/microorganisms12020365_

Round 1

Reviewer 1 Report

Comments and Suggestions for Authors

The article is accurate, the statistical analysis extensive and satisfactory, the conclusions are not very accurate.

Comments on the Quality of English Language

English is fluent and understandable even without advanced knowledge.

Reviewer 2 Report

Comments and Suggestions for Authors

The present study investigated the prevalence of anti-HEV among indigenous and non-indigenous populations living in the Brazilian Amazon basin. The results showed that overall prevalence of anti-HEV was 6.8% with 2.8% found in Yanomami areas, 3% in an urban area and 14.2% in a rural area. Although the study provided valuable data on the seroprevalence of hepatitis E virus in the Brazilian Amazon basin area, some major concerns exist and listed as below.

1. The surveys were conducted in the specific region and the findings provided limited insights for a broader area.

2. The sample size was insufficient to draw a conclusion on the prevalence of anti-HEV in the area.

3. The detection of anti-HEV IgG provided limited information for the prevalence of HEV infection. Further HEV RNA test are needed to verify the HEV infection in the investigated region.

4. The variables included in the logistic regression analysis were not comprehensive. Factors that are closely related to HEV infection should be added such as eating habits, source of water, etc.

5. Did the excluded subjects due to borderline or indeterminate anti-HEV results receive repeat tests for validation?

Comments on the Quality of English Language

Moderate editing of English language required.

Reviewer 3 Report

Comments and Suggestions for Authors

            The authors analyzed the seroprevalence of hepatitis E virus in different population groups from the Brazilian Amazon Basin. They found the higher prevalence in rural area. Some concerns should be addressed before publication.

1.     Page 2, Introduction first paragraph: The authors should include a best description of the 7-8 HEV genotypes, differentiating the zoonotic and the human ones. A good review of this is available (Pavio N et al. 2017. Recent knowledge on hepatitis E virus in Suidae reservoirs and transmission routes to human. Vet Res 48, 78). The human HEV genotype 1 has also been described in America. For a review see Viera-Segura O et al. Viruses 2023, 15(9), 1911.

2.     Page 2, line 85: Although HEV is certainly a neglected disease, it should be advisable not to say that it is the most neglected one. Just for comparison, a PubMed search of the term HEV arose 6628 entries (for a virus discovered in 1990), while the search for the term HDV arose only 2984 (discovered in 1977). Although the comparison is very approximate, these two viruses affecting the liver have been considered neglected and it would be difficult to categorize them as one more neglected than the other.

3.     Previous studies on HEV seroprevalence has been performed in Brazil, even in populations form the Amazon. The authors should stress the new findings added with their study.

4.     Page 4: Serological analysis. The authors do not explain in this section that one ELISA test or the other was used. This explanation is given as a limitation in the Discussion. Did the authors test a subset of their samples with both tests to compare their performance? That would be crucial to compare seroprevalence.

5.     Table 1: the significantly higher prevalence among male indigenous individuals is not mentioned further nor discussed.

6.     Page 8: Discussion, line 251. The terms HEV-1 and HEV-2 are misleading (see also lines 254 and 318 and Abstract). It would be better to the HEV genotypes as for example HEV G1 and G2.

7.     Page 8, line 251. HEV G2 is only found in highly endemic regions for HEV.

Round 2

Reviewer 3 Report

Comments and Suggestions for Authors

The authors addressed satisfactorely the concerns of the reviewers.

Author Response

Thanks for this comment. We have deleted this part. 
